# New Quinolone-Based Thiosemicarbazones Showing Activity Against *Plasmodium falciparum* and *Mycobacterium tuberculosis*

**DOI:** 10.3390/molecules24091740

**Published:** 2019-05-04

**Authors:** Richard M. Beteck, Ronnett Seldon, Audrey Jordaan, Digby F. Warner, Heinrich C. Hoppe, Dustin Laming, Setshaba D. Khanye

**Affiliations:** 1Division of Pharmaceutical Chemistry, Faculty of Pharmacy, Rhodes University, Grahamstown 6140, South Africa; 2Drug Discovery and Development Centre (H3-D), Department of Chemistry, University of Cape Town, Rondebosch 7701, South Africa; ronnett.seldon@uct.ac.za; 3SAMRC/NHLS/UCT Molecular Mycobacteriology Research Unit, Department of Pathology, University of Cape Town, Observatory 7925, South Africa; audrey.jordaan@uct.ac.za (A.J.); Digby.Warner@uct.ac.za (D.F.W.); 4Institute of Infectious Disease and Molecular Medicine, University of Cape Town, Observatory 7925, South Africa; 5Wellcome Centre for Infectious Diseases Research in Africa, University of Cape Town, Observatory 7925, South Africa; 6Department of Biochemistry and Microbiology, Faculty of Science, Rhodes University, Grahamstown 6140, South Africa; h.hoppe@ru.ac.za; 7Centre for Chemico- and Biomedicinal Research, Rhodes University, Grahamstown 6140, South Africa; dustinlaming89@gmail.com

**Keywords:** Co-infections, thiosemicarbazones, quinolones, malaria, *Mycobacterium tuberculosis*

## Abstract

Co-infection of malaria and tuberculosis, although not thoroughly investigated, has been noted. With the increasing prevalence of tuberculosis in the African region, wherein malaria is endemic, it is intuitive to suggest that the probability of co-infection with these diseases is likely to increase. To avoid the issue of drug-drug interactions when managing co-infections, it is imperative to investigate new molecules with dual activities against the causal agents of these diseases. To this effect, a small library of quinolone-thiosemicarbazones was synthesised and evaluated in vitro against *Plasmodium falciparum* and *Mycobacterium tuberculosis*, the causal agents of malaria and tuberculosis, respectively. The compounds were also evaluated against HeLa cells for overt cytotoxicity. Most compounds in this series exhibited activities against both organisms, with compound **10**, emerging as the hit; with an MIC_90_ of 2 µM against H37Rv strain of *M. tuberculosis* and an IC_50_ of 1 µM against the 3D7 strain of *P. falciparum*. This study highlights quinolone-thiosemicarabazones as a class of compounds that can be exploited further in search of novel, safe agents with potent activities against both the causal agents of malaria and tuberculosis.

## 1. Introduction

*Mycobacterium tuberculosis* (*Mtb*) is a pathogenic bacterium that causes tuberculosis, primarily a pulmonary infection in humans [1]. This bacterium currently infects about one-third of the world population [2]; however, a healthy immune system suppresses growth and multiplication of *Mtb* to an extent that the bacterium remains inactive and unable to cause pathology [3]. This is the case for 90% of people harbouring *Mtb* [4].

When the immune system becomes suppressed or weakened as is the case during treatment of rheumatoid arthritis, HIV infection, and old age, *Mtb* reactives, and consequently latent tuberculosis is converted to active pulmonary tuberculosis [5]. Other factors promoting conversion to the active state include diabetes and malnutrition [6]. People with active pulmonary tuberculosis suffer from symptoms of the disease [7] and can infect others [8]. An estimated 10.0 million people worldwide were reported to be infected by active pulmonary tuberculosis, and 1.3 million people died as consequence in 2017 [9]. The disease is currently treated using a first line regiment of four drugs (rifampicin, isoniazid, pyrazinamide, ethambutol), which must be taken daily for at least six months [10]. The current management and/or treatment of tuberculosis is complicated, and the situation has been worsened by the emergence and spread of multi-drug resistant and extensively drug resistant forms of *Mtb* strains [11]. There is therefore a great need to research new chemical agents with the potential to inhibit *Mtb* infection.

On the other hand, malaria is currently the third leading cause of deaths from a single infective agent. It caused ill health in 216 million people, and reportedly led to the death of 445,000 people in 2016 [12]. The disease is the biggest burden to Sub-Sahara Africa, wherein 88% of malarial cases and 90% of the reported deaths occurred [13]. *Plasmodium falciparum* is one of the five *Plasmodium* species responsible for malaria infection in humans [14]. It is the most virulent of all species [15] and accounts for 90 – 99% of the global malaria cases [16]. Although the death toll due to malaria has witnessed a decline over the past fifteen years [17], malaria is still a great global health challenge, with at least 40% of the world‘s population currently living in places endemic to this disease [18] and 90 countries still having ongoing malaria transmission [16]. The current scenario of malaria is worrisome, and is characterised by the emergence and spread of parasites resistant to all current treatment options, including artemisinins, the current treatment gold standard [19,20]. Artemisinin combination therapy (ACTs) is the recommended treatment for uncomplicated malaria [21], while quinine or artesunate administered parenterally is recommended for the treatment of complicated malaria [22].

A great common trait between malaria and tuberculosis is the development of resistance. A careful look into the history of malarial chemotherapy and treatment of tuberculosis suggests that the pathogens responsible for these diseases have an inherent ability to develop resistance over any drug used against them, irrespective of whether the drug(s) is used in mono- or combination therapy [23,24]. It is noteworthy to emphasise that these two diseases share common endemicity and the issue of co-infection with both diseases has been reported [25]. With increasing prevalence of TB in the African region (the area with the highest malaria prevalence), there is likely to be a high prevalence of TB and malaria co-infection. Furthermore, it has been noted that malaria causes a deterioration in TB treatment outcome [25] and commonly deployed anti-malarials such as chloroquine and quinine are counter indicative with fluoroquinolones, second line anti-TB agents [26]. Taking these in consideration, it should be intuitive to search for new molecules with the potential to act as anti-malarial and anti-TB agents.

Thioacetazone (**1**, Figure 1), a thiosemicarbazone-containing compound, is a potent anti-TB drug. Inherent toxicities issues associated with the use of this drug especially in situations of HIV infection, a common TB co-infection, has jeopardised further clinical deployment of this drug [27]. Other thiosemicarbazone containing compounds (**2**, Figure 1) have been reported to exhibit anti-malarial activity [28]. Fluoroquinolones (**3**, Figure 1), a sub-class of quinolones, are recommended second line therapy for TB [29]. More importantly, fluoroquinolones have also been reported for their anti-malarial potential (**4**, Figure 1) [30]. In recent work, we reported quinolones incorporating a hydrazide-hydrazone moiety (**5**, Figure 1), which exhibited potent activity against *Mtb* [31]. Considering these literatures precedents, we reasoned that design and development of novel compounds incorporating both thiosemicarbazone and quinolone frameworks might be worth exploiting as potential compounds to target both malaria and TB infections. Herein, we report the synthesis of heterocyclic compounds with thiosemicarbazone and quinolone motifs at their core structure. In vitro biological evaluations showcase these compounds as potent inhibitors of both *P. falciparum* and *M. tuberculosis*.

## 2. Results and Discussion

### 2.1. Chemistry

Our target compounds were obtained following the synthetic routes presented in Scheme 1 below. Briefly, the reduction of *p*-nitroacetophenone **6** using reduced iron powder and ammonium chloride in refluxing ethanol afforded *p*-aminoacetophenone **7**. Compound **7** was condensed with diethyl ethoxy methylenemalonate, and following a classical Gould-Jacobs cyclization procedure for quinolone synthesis [32], later cyclised in boiling diphenylether to obtain intermediate **8** in 50–70% yields. Compound **8**, just like most quinolones reported in literature, is practically not soluble in any solvent [32]; this partly could be attributed to *pi*-*pi* stacking promoted by the highly planar nature of this fragment in particular, and the quinolone nucleus in general. It was therefore essential to modify this fragment in such a way as to disrupt planarity, and/or introduce hydrophilic moieties before appending a thiosemicarbazone unit, this was achieved through *N*-alkylation of the secondary amine and aminolysis of the ester functionalities in **8**. *N*-alkylation of the secondary amine was achieved following treatment with alkyl/aryl halides under refluxing DMF using K_2_CO_3_ as a base to afford intermediate **9** in 50–70% yields. Compound **9** was subjected to aminolysis by treatment with hydrophilic amines in the presence of DBU under refluxing conditions to obtain **10** as amide derivative in 40% yield. Treatment of **9**, or **10** with equimolar amounts of thiosemicarbazide and a few drops of acetic acid gave target compounds **11**–**21** in 60–80% yields. The structures of target compounds were confirmed using ^1^H and ^13^C NMR, high resolution mass spectrometry analysis also confirmed the pseudomolecular ion peaks consistent with structures of achieved compounds. (See Appendix A). The signal appearing at *ca* 10.11 ppm on the ^1^H NMR spectra of compound **11**–**19** is indicative of the presence of an amide proton (-CONH-), this signal is absent in the ^1^H NMR spectra of compound **20** and **21**, both of which lack the corresponding amide. The peak at *ca* 10.37 ppm on the ^1^H NMR spectra of all target compounds is attributable to the hydrazinic proton (-C(=S)NH-N=). The thioamide protons (-C(=S)NH_2_) could be observed at two different chemical shifts, *ca* 8.3 ppm and 8.00 ppm. These differences in chemical shift to protons attached to the same atom have been attributed to the lack of free rotation brought about by the formation of carbon-nitrogen double bond character around C-N bond of the thioamide [33].

### 2.2. Pharmacology

All target compounds were screened for in vitro inhibitory activity against the 3D7 strain of *P. falciparum* and the *M. tuberculosis* H37Rv strain. Target compounds were also evaluated in vitro against human cervix adernocarcinoma (HeLa) cell for overt cytotoxicity. Emetine, a cytotoxic agent, was used as a reference in the latter assay. At 20 µM, none of the compounds suppressed HeLa cell viability below 80%, suggesting that they pose little cytotoxicity risk at this concentration. Target compounds were evaluated for their potential to inhibit the growth of *M. tuberculosis* in vitro using middlebrook 7H9 media supplemented with casitone, glucose and tyloxpol [34]. The H37Rv strain of *Mtb* was deployed in this study, and rifampicin was used as a reference. The anti-*Mtb* activity of compounds in this study is reported as the minimum inhibitory concentration required to inhibit 90% (MIC_90_) of the bacteria population and the data is summarised in Table 1 below. Except compound **11**, all compounds were active against *Mtb*; exhibiting activity in the range of 2–100 µM.

The anti-*Mtb* potential of this series appeared to be strongly influenced by the nature of substituents at position -1 and -3 of the quinolone ring. For-example, comparing compound **20** (MIC_90_; 102.5 µM) with compound **14** (MIC_90_; 10.3 µM) indicates that an amide at position -3 promotes activity over an ester group. This observation is also evident when comparing compound **21** (MIC_90_; 31.6 µM) with compound **16** (MIC_90_; 4.8 µM). Structure-activity relationship analysis further suggests that activity increases with increasing hydrophilicity of the amide moiety at position -3. This is observed when comparing the structure and activity of, for example, compound **17** (MIC_90_; 44.2 µM) with compound **13** (MIC_90_; 2.7 µM), compound **11** (MIC_90_; ˃125 µM) versus compound **12** (MIC_90_; 73.4 µM). Moreover, comparing the structure and activity of compound **13** (MIC_90_; 2.7 µM) with compound **12** (MIC_90_; 73.4 µM) highlights that the presence of benzyl unit at position -1 seems to be favoured over an alkyl chain. There, is however, lack of a clear correlation of how the substitution pattern on the benzyl unit influences activity.

To investigate anti-plasmodial activity, target compounds were screened in vitro against the chloroquine sensitive strain (3D7) of *P. falciparum* in a pLDH assay using chloroquine as the reference. Activity of these compounds together with the reference compound is reported as IC_50_, concentration required to inhibit 50% parasite growth and is incorporated in Table 1. Most compounds in this series exhibited activity in the low micromolar range (IC_50_ less than 5 µM). The anti-plasmodial activity of this series follows the same pattern noted for anti-*Mtb* activity. For example, comparing compound **12** (not active against 3D7) and **8** (IC_50_; 4 µM), both of which differ only on the nature of substituent at position -1, suggest that a benzyl unit at this position leads to favourable anti-plasmodial activity. Compound **20**, bearing an ethyl ester at position -3, is devoid of activity, while compound **14**, its amide derivative, exhibited an IC_50_ value of 4 µM. This observation points to the fact that an amide at this position greatly enhances activity over an ester moiety.

## 3. Materials and Methods

### 3.1. General Information

Chemicals and solvents deployed in this study were purchased from various chemical vendors: Sigma-Aldrich (Pty) Ltd. (Johannesburg, South Africa), Merck (Pty) Ltd. (Johannesburg, South Africa) and were used as purchased. The progress of the reactions was monitored by thin layer chromatography (TLC) using Merck 60F254 silica gel plates (Merck, Johannesburg, South Africa) supported on aluminum and the plates were visualized under ultraviolet (UV 254 and 366 nm) light and in iodine flasks. Where necessary, the crude compounds were purified by means of a silica gel column chromatography using Merck Kieselgel 60 Å: 70–230 (0.068–0.2 mm) silica gel mesh (Merck, Johannesburg, South Africa). ^1^H and ^13^C NMR spectra were recorded on Bruker Biospin 300 MHz spectrometer, and the chemical shifts (δ) are given in values referenced to deuterated DMSO-*d_6_* and are reported in parts per million (ppm). Proton chemical shifts for deuterated DMSO appear at 2.5 ppm for ^1^H and 39.5 ppm for ^13^C-NMR. Proton coupling patterns are abbreviated as follows: s (singlet), d (doublet), t (triplet), q (quartet), and m (multiplet). Coupling constant (*J*) are reported in Hz. The high-resolution mass spectrometric data (HRS-MS) of final compounds were recorded on Waters Synapt G2 Mass Spectrometer (Stellenbosch University, South Africa) using electrospray ionization (ESI) in the positive ionization mode. Melting points were obtained using a Reichert hot stage microscope and are reported as obtained. The starting compounds **7–10** were synthesized from the commercial accessible *p*-nitroacetophenone **6** as previously described in the literature [34]. Purity was determined by HPLC (Agilent, Santa Clara, CA, USA), and all compounds were confirmed to have purity >95% using a similar method previously described [35].

### 3.2. General Synthetic Procedure for the Quinolone-Thiosemicarbazone Derivatives, 11–21

In a round bottom flask containing 400 mg of compound **9** or **10** dissolved in 95% ethanol (20 mL) was added 1.2 equivalence of thiosemicarbazide followed by a few drops of glacial acetic acid. The reaction mixture was stirred under reflux overnight, left to cool to room temperature. The formed precipitates were filtered, washed with 2 × 10 mL ice cold ethanol and air dried to obtain target compounds, **11 – 21**, in 60–80% yields.

(*E)-N-(3-(1H-imidazol-1-yl)propyl)-6-(1-(2-carbamothioylhydrazono)ethyl)-1-ethyl-4-oxo-1,4-dihydroquinolone-3-carboxamide*, **11**. Brown powder. Yield = 80% (0.35 g). m.p.: 187–189 °C. ^1^H-NMR (300 MHz, DMSO-*d_6_*) δ_H_ (ppm): 10.37 (s, 1H, CSNHN), 10.04 (t, *J* = 5.8 Hz, 1H, CONH), 8.86 (s, 1H, Ar-H), 8.69 – 8.47 (m, 3H, Ar-H), 8.36 (s, 1H, NH), 8.03 (s, 1H, NH), 7.83 (d, *J* = 9.2 Hz, 1H, Ar-H), 7.68 (s, 1H, Ar-H), 7.22 (s, 1H, Ar-H), 4.53 (q, *J* = 6.9 Hz, 2H, -CH_2_CH_3_), 4.03 (t, *J* = 6.9 Hz, 2H, -CH_2_-), 3.60–3.29 (m, 6H, 2 × -CH_2,_ overlap with *d*H_2_O), 2.40 (s, 3H, CH_3_), 1.39 (t, *J* = 6.9 Hz, 3H, -CH_2_CH_3_). ^13^C NMR (75 MHz, DMSO-*d_6_*) δ_C_ (ppm): 179.9, 173.1, 165.4, 148.1, 146.4, 140.0, 137.4, 134.6, 131.4, 128.3, 126.9, 124.7, 120.1, 117.9, 111.9, 56.3, 48.2, 44.6, 35.5, 15.2, 14.2. **m*/*z** (ESI-MS) found 440.1860, calcd for C_21_H_26_N_7_O_2_S 440.1869 [M + H]^+^. HPLC purity > 96%, rt = 8.2 min.

*(E)-6-(1-(2-carbamothioylhydrazono)ethyl)-1-ethyl-N-(2-((2-hydroxyethyl)amino)ethyl)-4-oxo-1,4-dihydroquinolone-3-carboxamide*, **12**. Yellow powder. Yield = 60% (0.2 g). m.p.: 183–185 °C. ^1^H-NMR (300 MHz, DMSO-d_6_) δ_H_ (ppm): 10.35 (s, 1H, CSNHN), 10.12 (t, J = 4.9 Hz, 1H, CONH), 8.86 (s, 1H, Ar-H), 8.69 – 8.47 (m, 2H, Ar-H), 8.36 (s, 1H, NH), 8.03 (s, 1H, NH), 7.83 (d, J = 8.9 Hz, 1H, Ar-H), 5.31 (s, 1H, OH), 4.53 (q, J = 6.8 Hz, 2H, -CH_2_CH_3_), 3.74 – 3.54 (m, 5H, -NH-, 2 × -CH_2_-), 3.21–2.95 (m, 4H, 2 × -CH_2_-), 2.39 (s, 3H, CH_3_), 1.43 (t, J = 6.8 Hz, 3H, -CH_2_CH_3_). ^13^C-NMR (75 MHz, DMSO-d_6_) δ_C_ (ppm): 179.4, 175.4, 165.9, 148.7, 146.8, 140.0, 135.1, 131.9, 127.9, 124.7, 117.9, 111.6, 56.8, 49.5, 49.1, 46.9, 35.9, 15.1, 14.2. *m*/*z* (ESI-MS) found 419.1971, calcd for C_19_H_27_N_6_O_3_S 419.1966 [M + H]^+^. HPLC purity > 96%, rt = 6.4 min.

*(E)-1-benzyl-6-(1-(2-carbamothioylhydrazono)ethyl)-N-(2-((2-hydroxyethyl)amino)ethyl)-4-oxo-1,4-dihydroquinolone-3-carboxamide*, **13**. White powder. Yield = 70% (0.34 g). m.p.: 197–199 °C. ^1^H-NMR (300 MHz, DMSO-d_6_) δ_H_ (ppm): 10.35 (s, 1H, CSNHN), 10.10 (t, J = 5.6 Hz, 1H, CONH), 9.08 (s, 1H, Ar-H), 8.64 – 8.44 (m, 2H, Ar-H), 8.32 (s, 1H, NH), 8.02 (s, 1H, NH), 7.67 (d, J = 9.2 Hz, 1H, Ar-H), 7.47 – 7.10 (m, 5H, Ar-H), 5.83 (s, 2H, -CH_2_Ar), 5.30 (s, 1H, OH), 3.94 – 3.51 (m, 5H, -NH-, 2 × -CH_2_-), 3.16 (t, J = 5.6 Hz, 2H, -CH_2_-), 3.05 (t, J = 6.7 Hz, 2H, -CH_2_-), 2.36 (s, 3H, -CH_3_). ^13^C-NMR (75 MHz, DMSO-d_6_) δ_C_ (ppm): 179.9, 175.8, 165.9, 149.1, 146.4, 139.9, 136.8, 135.5, 131.4, 129.6, 129.2, 128.3, 127.9, 127.4, 126.9, 124.22, 119.2, 111.9, 56.8, 56.3, 49.9, 46.8, 35.9, 14.2. *m*/*z* (ESI-MS) found 481.1930, calcd for C_24_H_29_N_6_O_3_S 481.1925 [M+H]^+^. HPLC purity > 96%, rt = 8.5 min.

*(E)-1-(4-bromobenzyl)-6-(1-(2-carbamothioylhydrazono)ethyl)-N-(2-((2-hydroxyethyl)amino)ethyl)-4-oxo-1,4-dihydroquinolone-3-carboxamide*, **14**. White powder. Yield = 80% (0.34 g). m.p.: 200–202 °C. ^1^H-NMR (300 MHz, DMSO-d_6_) δ_H_ (ppm): 10.36 (s, 1H, CSNHN), 10.11 (t, J = 5.8 Hz, 1H, CONH), 9.08 (s, 1H, Ar-H), 8.61 – 8.44 (m, 2H, Ar-H), 8.33 (s, 1H, NH), 8.01 (s, 1H, NH), 7.64 (d, J = 9.8 Hz, 1H, Ar-H), 7.55 (d, J = 8.4 Hz, 2H, Ar-H), 7.19 (d, J = 8.4 Hz, 2H, Ar-H), 5.82 (s, 2H, -CH_2_Ar), 5.27 (s, 1H, OH), 3.77 – 3.63 (m, 5H, -NH-, 2 × -CH_2_-), 3.16 (t, J = 6.1 Hz, 2H, -CH_2_-), 3.09 – 3.00 (m, 2H, -CH_2_-), 2.36 (s, 3H, -CH_3_). ^13^C-NMR (75 MHz, DMSO-d_6_) δ_H_ (ppm): 179.5, 175.4, 165.4, 146.4, 144.1, 140.0, 136.4, 135.1, 134.6, 132.4, 129.6, 127.9, 124.2, 121.9, 118.3, 111.9, 56.3, 55.4, 50.0, 47.2, 35.9, 14.2. *m*/*z* (ESI-MS) found 559.1123, calcd for C_24_H_28_BrN_6_O_3_S 559.1127 [M + H]^+^. HPLC purity > 96%, rt = 9.4 min.

*(E)-6-(1-(2-carbamothioylhydrazono)ethyl)-N-(2-((2-hydroxyethyl)amino)ethyl)-4-oxo-1-(4-(trifluoromethyl)benzyl)-1,4-dihydroquinolone-3-carboxamide*, **15**. White powder. Yield = 80% (0.35 g). m.p.: 199–201 °C. ^1^H-NMR (300 MHz, DMSO-d_6_) δ_H_ (ppm): 10.37 (s, 1H, CSNHN), 10.11 (s, 1H, CONH), 9.12 (s, 1H, Ar-H), 8.56 (s, 1H, Ar-H), 8.32 (s, 1H, CSNH), 8.01 (s, 1H, CSNH), 7.87 – 7.27 (m, 6H, Ar-H), 5.97 (s, 2H, -CH_2_Ar), 5.26 (s, 1H, OH), 3.79 – 3.50 (m, 4H, 2 × -CH_2_), 3.10 – 2.87 (m, 5H, -NH-, 2 × -CH_2_), 2.36 (s, 3H, -CH_3_). ^13^C-NMR (75 MHz, DMSO-d_6_) δ_C_ (ppm): 179.9, 175.8, 165.9, 149.6, 146.4, 141.4, 140.5, 135.5, 135.0, 131.7, 127.6, 127.5, 126.3, 124.7, 122.9, 118.3, 112.5, 56.8, 55.4, 49.5, 47.3, 35.9, 14.2. *m*/*z* (ESI-MS) found 549.1790, calcd for C_25_H_28_F_3_N_6_O_3_S 549.0896 [M + H]^+^. HPLC purity > 96%, rt = 9.4 min.

*(E)-6-(1-(2-carbamothioylhydrazono)ethyl)-1-(2,4-dichlorobenzyl)-N-(2-((2-hydroxyethyl)amino)ethyl)-4-oxo-1,4-dihydroquinolone-3-carboxamide*, **16**. Off white powder. Yield = 80% (0.32 g). m.p.: 205–207 °C. ^1^H-NMR (300 MHz, DMSO-d_6_) δ_H_ (ppm): 10.39 (s, 1H, CSNHN), 10.10 (t, J = 5.0 Hz, 1H, CONH), 9.12 (s, 1H, Ar-H), 8.69 – 8.47 (m, 2H, Ar-H), 8.36 (s, 1H, CSNH), 8.05 (s, 1H, CSNH), 7.78 – 7.50 (m, 3H, Ar-H), 7.15 (dd, J = 7.8, 5.8 Hz, 1H, Ar-H), 5.85 (s, 2H, -CH_2_Ar), 5.32 (t, J = 4.7 Hz, 1H, OH), 3.92 – 3.54 (m, 5H, -NH-, 2 × -CH_2_-), 3.16 (t, J = 5.9 Hz, 2H, -CH_2_-), 3.05 (t, J = 5.0 Hz, 2H, -CH_2_-), 2.36 (s, 3H, -CH_3_). ^13^C-NMR (75 MHz, DMSO-d_6_) δ_C_ (ppm): 179.4, 176.3, 165.4, 149.9, 146.4, 145.9, 139.6, 137.7, 135.5, 132.4, 131.9, 131.4, 131.0, 129.6, 127.4, 124.7, 118.8, 111.9, 57.2, 55.4, 49.5, 46.8, 35.9, 14.2. *m*/*z* (ESI-MS) found 549.1126, calcd for C_24_H_27_Cl_2_N_6_O_3_S 549.1119 [M + H]^+^. HPLC purity > 96%, rt = 9.1 min.

*(E)-1-benzyl-6-(1-(2-carbamothioylhydrazono)ethyl)-N-(2-methoxyethyl)-4-oxo-1,4-dihydroquinolone-3-carboxamide,***17**. Off white powder. Yield = 73% (0.34 g). m.p.: 199–200 °C. ^1^H-NMR (300 MHz, DMSO-*d_6_*) δ_H_ (ppm): 10.43 (s, 1H, CSNHN), 10.17 (t, *J* = 5.0 Hz, 1H, CONH), 9.14 (s, 1H, Ar-H), 8.70 – 8.50 (m, 2H, Ar-H), 8.40 (s, 1H, NH), 8.11 (s, 1H, NH), 7.73 (d, *J* = 8.7 Hz, 1H, Ar-H), 7.48 – 7.16 (m, 5H, Ar-H), 5.88 (s, 2H, -CH_2_Ar), 3.66 – 3.52 (m, 4H, 2 × -CH_2_-), 3.37 (s, 3H, OCH_3_), 2.42 (s, 3H, CH_3_). ^13^C-NMR (75 MHz, DMSO-*d_6_*) δ_C_ (ppm): 179.9, 175.8, 164.4, 146.8, 140.5, 136.8, 134.6, 131.9, 130.1, 129.6, 129.2, 128.3, 127.9, 127.4, 126.9, 125.1, 118.3, 111.6, 71.7, 58.5, 57.2, 38.7, 14.2. **m*/*z** (ESI-MS) found 452.1660, calcd for C_23_H_26_N_5_O_3_S 452.0981 [M + H]^+^. HPLC purity > 96%, rt = 9 min.

*(E)-1-(4-bromobenzyl)-6-(1-(2-carbamothioylhydrazono)ethyl)-N-(2-methoxyethyl)-4-oxo-1,4-dihydroquinolone-3-carboxamide,***18**. Brown powder. Yield = 60% (0.25 g). m.p.: 204–206 °C. ^1^H-NMR (300 MHz, DMSO-*d_6_*) δ_H_ (ppm): 10.34 (s, 1H, CSNHN), 10.07 (t, *J* = 5.1 Hz, 1H, CONH), 9.09 (s, 1H, Ar-H), 8.64 – 8.47 (m, 2H, Ar-H), 8.31 (s, 1H, NH), 8.02 (s, 1H, NH), 7.64 – 7.44 (m, 3H, Ar-H), 7.20 (d, *J* = 8.9 Hz, 2H, Ar-H), 5.80 (s, 2H, -CH_2_Ar), 3.60 – 3.46 (m, 4H, 2 × -CH_2_-), 3.26 (s, 3H, OCH_3_), 2.36 (s, 3H, CH_3_). ^13^C-NMR (75 MHz, DMSO-*d_6_*) δ_C_ (ppm): 180.4, 175.8, 164.5, 149.6, 146.4, 139.6, 136.5, 135.1, 132.4, 131.5, 129.2, 127.4, 124.7, 121.5, 118.8, 111.9, 71.3, 58.5, 55.9, 38.7, 13.8. **m*/*z** (ESI-MS) found 530.0847, calcd for C_23_H_25_BrN_5_O_3_S 530.0861 [M + H]^+^. HPLC purity > 96%, rt = 10.5 min.

*(E)-6-(1-(2-carbamothioylhydrazono)ethyl)-1-(2,4-dichlorobenzyl)-N-(2-methoxyethyl)-4-oxo-1,4-dihydroquinolone-3-carboxamide*, **19**. Brown powder. Yield = 65% (0.27 g). m.p.: 209–211 °C. ^1^H-NMR (300 MHz, DMSO-d_6_) δ_H_ (ppm): 10.34 (s, 1H, CSNHN), 10.05 (s, 1H, CONH), 9.07 (s, 1H, Ar-H), 8.64 – 8.47 (m, 2H, Ar-H), 8.35 (s, 1H, NH), 8.03 (s, 1H, NH), 7.73 – 7.44 (m, 3H, Ar-H), 7.18 (s, 1H, Ar-H), 5.82 (s, 2H, -CH_2_Ar), 3.66 – 3.46 (m, 6H, 2 × -CH_2_-, overlap with H_2_O), 3.31 (s, 3H, -OCH_3_), 2.37 (s, 3H, CH_3_). ^13^C NMR (75 MHz, DMSO-d_6_) δ 179.5, 176.7, 164.1, 146.8, 144.6, 139.9, 138.3, 135.1, 134.6, 132.8, 131.9, 130.5, 129.6, 127.9, 127.4, 124.7, 118.8, 112.5, 71.3, 62.6, 58.5, 39.1, 14.7. *m*/*z* (ESI-MS): found 520.0857, calcd for C_23_H_24_Cl_2_N_5_O_3_S 520.0861 [M + H]^+^. HPLC purity > 96%, rt = 9.4 min.

*Methyl-(E)-1-(4-bromobenzyl)-6-(1-(2-carbamothioylhydrazono)ethyl)-4-oxo-1,4-dihydroquinolone-3-carboxylate,***20**. Off white powder. Yield = 60% (0.25 g). m.p.: 203–205 °C. ^1^H-NMR (300 MHz, DMSO-*d_6_*) δ_H_ (ppm): 10.31 (s, 1H, CSNHN), 8.94 (s, 1H, Ar-H), 8.52 – 8.38 (m, 3H, Ar-H), 8.30 (s, 1H, NH), 7.99 (s, 1H, NH), 7.55 (d, *J* = 8.7 Hz, 2H, Ar-H), 7.23 (d, *J* = 8.7 Hz, 2H, Ar-H), 5.69 (s, 2H, -CH_2_Ar), 3.78 (s, 3H, -OCH_3_), 2.34 (s, 3H, -CH_3_). ^13^C-NMR (75 MHz, DMSO-*d_6_*) δ_C_ (ppm): 179.5, 173.5, 164.9, 150.5, 146.4, 139.2, 135.9, 135.1, 132.3, 131.05, 129.6, 128.3, 124.7, 121.4, 118.8, 111.0, 55.0, 51.8, 14.7. **m*/*z** (ESI-MS) found 487.0437, calcd for C_21_H_20_BrN_4_O_3_S 487.0439 [M + H]^+^. HPLC purity > 96%, rt = 8.5 min.

*Ethyl-(E)-6-(1-(2-carbamothioylhydrazono)ethyl)-1-(2,4-dichlorobenzyl)-4-oxo-1,4-dihydroquinolone-3-carboxylate,***21**. Brown powder. Yield = 60% (0.23 g). m.p.: 211–213 °C. ^1^H-NMR (300 MHz, DMSO-*d_6_*) δ_H_ (ppm): 10.35 (s, 1H, CSNHN), 8.92 (s, 1H, Ar-H), 8.58 – 8.41 (m, 2H, Ar-H), 8.33 (s, 1H, NH), 8.03 (s, 1H, NH), 7.75 – 7.38 (m, 3H, Ar-H), 7.17 (d, *J* = 9.0 Hz, 1H, Ar-H), 5.72 (s, 2H, -CH_2_Ar), 4.24 (q, *J* = 6.7 Hz, 2H, -OCH_2_-), 2.34 (s, 3H, CH_3_), 1.30 (t, *J* = 6.7 Hz, 3H, -CH_3_). ^13^C-NMR (75 MHz, DMSO-*d_6_*) δ_C_ (ppm): 179.5, 173.2, 164.9, 150.5, 146.4, 140.5, 136.4, 134.6, 133.7, 132.8, 131.9, 129.6, 128.7, 127.9, 126.9, 124.2, 118.4, 111.6, 60.4, 54.9, 15.5, 14.2. **m*/*z** (ESI-MS) found 491.0610, calcd for C_22_H_21_Cl_2_N_4_O_3_S 491.0615 [M + H]^+^. HPLC purity > 96%, rt = 9.4 min.

### 3.3. In Vitro Anti-Plasmodial Assay

The 3D7 strain of *Plasmodium falciparum* was routinely cultured in medium consisting of RPMI1640 containing 25 mM Hepes (Lonza), supplemented with 0.5% (w/v) Albumax II (ThermoScientific), 22 mM glucose, 0.65 mM hypoxanthine, 0.05 mg/mL gentamicin and 2–4% (v/v) human erythrocytes. Cultures were maintained at 37 °C under an atmosphere of 5% CO_2_, 5% O_2_, 90% N_2_. To assess antiplasmodial activity, three-fold serial dilutions of test compounds in culture medium were added to parasite cultures (adjusted to 2% parasitaemia, 1% haematocrit) in 96-well plates and incubated for 48 h. Duplicate wells per compound concentration were used. Parasite lactate dehydrogenase (pLDH) enzyme activity in the individual wells was determined as previously described [35,36].

### 3.4. In Vitro Cytotoxicity Assay

As previously described [37], HeLa cells (Cellonex) seeded in 96-well plates were incubated with 20 µM test compounds for 24 h and cell viability assessed using a resazurin fluorescence assay.

### 3.5. In Vitro Antimycobacterial Assay

The minimum inhibitory concentration (MIC) was determined using the standard broth microdilution method, where a 10 mL culture of *Mycobacterium tuberculosis* pMSp12::GFP [38], was grown to an optical density (OD600) of 0.6–0.7. The medium used was Middlebrook 7H9 supplemented with 0.03% casitone, 0.4% glucose, and 0.05% tyloxapol [39]. Cultures grown in the medium were diluted 1:500, prior to inoculation in the MIC assay. The compounds to be tested were reconstituted to a concentration of 10 mM in DMSO. Two-fold serial dilutions of the test compound were prepared across a 96-well micro titre plate, after which, 50 μL of the diluted *Mtb* cultures was added to each well in the serial dilution. Assay controls used were a minimum growth control (Rifampicin at 2 × MIC), and a maximum growth control (5% DMSO). The micro titre plates were sealed in a secondary container and incubated at 37 °C with 5% CO_2_ and humidification. Relative fluorescence (excitation 485 nM; emission 520 nM) was measured using a plate reader (FLUOstar OPTIMA, BMG LABTECH), at day 7 and day 14. The raw fluorescence data were archived and analysed using the CDD Vault from Collaborative Drug Discovery, in which data were normalised to the minimum and maximum inhibition controls to generate a dose response curve (% inhibition), using the Levenberg-Marquardt damped least squares method, from which the MIC_90_ was calculated (Burlingame, CA www.collaborativedrug.com). The lowest concentration of drug that inhibited growth of more than 90% of the bacterial population was considered the MIC_90_.

## 4. Conclusions

In this study, quinolone-thiosemicarbazones were obtained in high yields through simple and cost effective classical synthetic transformations without the use of specialised equipment and expensive catalysts. The compounds reported herein showed no growth inhibition of HeLa cells at a concentration of 20 µM suggesting the general lack of cytotoxicity by this series. More importantly, this series exhibited good activities against *P. falciparum*, and *M. tuberculosis*, the pathogens responsible for malaria, and tuberculosis, respectively. Overall, the presented data suggest that quinolone-thiosemicarbazones are a compound class worth further exploitation in search of cheap molecules with dual activities against the aforementioned diseases-causing microorganisms.

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
