# Peer review of "New Quinolone-Based Thiosemicarbazones Showing Activity Against Plasmodium falciparum and Mycobacterium tuberculosis"

_molecules, 2019, doi:10.3390/molecules24091740_

Round 1

Reviewer 1 Report

The chemical part is good. The results of biological tests are too weak compared to the standards used. Why did the Authors use cancer HeLa cell line instead of normal cell line to investigate the cytotoxicity of compounds?

Author Response

The chemical part is good. The results of biological tests are too weak compared to the standards used. Why did the Authors use cancer HeLa cell line instead of normal cell line to investigate the cytotoxicity of compounds?

We thank the reviewer for valuable comments. In response, we will  argue that in terms of TB any molecule with an MIC90 £ 10 µM is considered an “active” and is  worth pursuing for Mtb evaluation. Three compounds from this series showed significant activity against Mtb H37Rv strain despite the fact standard is  highly active. Similarly, while majority of these compounds week activity against the 3D7 (chloroquine-sensitive) strain of P. falciparum parasite at least four compounds proved active.

We do appreciate the question raised by reviewer #1. In response, for example, vero cells were derived from the kidney epithelial cells of an African green monkey. They are rapidly growing immortal cells, and it is      thus questionable that they can be considered “normal” (as opposed to, for example, primary cells). HeLa cells are human epithelial cells derived from a cervical cancer. As such, they are also rapidly growing, which means they are more susceptible to cytotoxicity than more slowly growing cell lines and can thus flag compounds that may have potentially debilitating effects against human cells. They are the most commonly used cells in research, including wide use for cytotoxicity assays (compared to, for example, Vero cells). This means our results may be more directly comparable to results obtained with other compound classes in drug discovery projects.

Reviewer 2 Report

The manuscript needs minor changes in the text as indicated in the attached file.

The conclusion part needs to be written well. I am not sure if it is summary with a heading as "Conclusion" and after few lines it is mentioned as summary.

Author Response

“The manuscript needs minor changes in the text as indicated in the attached file. The conclusion part needs to be written well. I am not sure if it is summary with a heading as "Conclusion" and after few lines it is mentioned as summary.”

We thank the reviewer #2 for feedback. The changes in the text including incorporation of most recent reference(s) have been done as suggested. We have also updated the conclusion section to eliminate ambiguity and confusion.

Round 2

Reviewer 1 Report

Accept